# Complexity and Community Context: Learning from the Evaluation Design of a National Community Empowerment Programme

**DOI:** 10.3390/ijerph17010091

**Published:** 2019-12-21

**Authors:** Jane South, Daniel Button, Annie Quick, Anne-Marie Bagnall, Joanne Trigwell, Jenny Woodward, Susan Coan, Kris Southby

**Affiliations:** 1Centre for Health Promotion Research, School of Health and Community Studies, Leeds Beckett University, Leeds LS1 3HE, UK; A.Bagnall@leedsbeckett.ac.uk (A.-M.B.); j.trigwell@leedsbeckett.ac.uk (J.T.); J.L.Woodward@leedsbeckett.ac.uk (J.W.); S.Coan@leedsbeckett.ac.uk (S.C.); K.Southby@leedsbeckett.ac.uk (K.S.); 2New Economics Foundation, London SE1 7HB, UK; Daniel.Button@neweconomics.org (D.B.); anniequicktalk@gmail.com (A.Q.)

**Keywords:** empowerment, community-based research, evaluation, case studies, complexity, neighbourhood deprivation

## Abstract

Community empowerment interventions, which aim to build greater individual and community control over health, are shaped by the community systems in which they are implemented. Drawing on complex systems thinking in public health research, this paper discusses the evaluation approach used for a UK community empowerment programme focused on disadvantaged neighbourhoods. It explores design choices and the tension between the overall enquiry questions, which were based on a programme theory of change, and the varied dynamic socio-cultural contexts in intervention communities. The paper concludes that the complexity of community systems needs to be accounted for through in-depth case studies that incorporate community perspectives.

## 1. Introduction

Interest in complex systems thinking in public health has been partly borne out of a frustration with the limitations of traditional evidence models [1,2] and an understanding that real world public health, which is conducted in the places where people live, work and play, needs multi-sectoral action to address the causes of the causes of poor health [3,4]. Despite wide acceptance of the need for a whole-of-government, whole-of-society approach to effect improvements in health and reduction of health inequalities [5,6], public health research methodologies that take account of system-wide action are relatively underdeveloped [7]. The dominant research paradigm tends to focus on defined individual-level interventions rather than exploring the levers for change that operate at different levels within a system, and the contribution of different actors, including communities [4,8]. Critique of the limitations of assessing change as a bounded, linear process has led to calls to apply systems thinking to public health evidence [1,2,9,10]. Moore and colleagues identify the areas where further methodological development is warranted. This includes evaluation methods for assessing effectiveness of “whole system changes which are not easily evaluable via a focus on discrete health outcomes” [11]. The frustration around an evidence-practice gap, where public health operates in multisectoral systems to effect change in local contexts, is particularly felt in relation to community empowerment interventions [12,13,14], as these interventions typically involve social action independent of the professional sphere and are non-standardised.

Community empowerment interventions aim to facilitate people’s active participation and to enable collective action that addresses health needs and inequities by changing community conditions [15,16]. Laverack and Pratley describe empowerment as a complex concept that is both multi-level and multi-dimensional [17]. The terms ‘empowerment’ and ‘control’ are often used interchangeably in the literature. Whitehead and colleagues describe collective or community control as going “beyond individual circumstances to encompass the strength/power generated by joining together to have greater influence over material and social conditions in immediate neighbourhood/living space” [18]. Within this wider determinants framework, increased control can be built at multiple levels: micro/personal; meso/community; macro/social [18]. The complexity of empowerment interventions not only relates to the multiple components of intervention and the ‘professional’ system that might support implementation, but also to the community systems where change is built as individual and collective agency grows [12,19,20]. There are debates over the best approach to measurement and what quantitative indicators are appropriate to assess empowerment processes and outcomes [13,17,21,22]. The traditional paradigm of public health evaluation, focused on the assessment of professionally-designed linear interventions using experimental designs, is generally ill-suited to evaluating developmental change processes in communities [23,24]. Raphael and Bryant highlight a further problem of ‘context stripping’, which occurs when population health research ignores the community and societal structures that impact on individuals’ health [25]. Exclusionary processes affecting the most marginalised communities interact with other determinants to result in a range of poor health and social outcomes [26,27]. Therefore, accounting for complexity in community-based research is an equity issue if the experiences of those most affected by inequalities are to be reflected in public health research.

Trickett and colleagues present an alternative paradigm for the study of community level interventions to address health and health inequalities based on an ecological/systems framework [24]. One of the central assertions is that community interventions should be conceptualised as a set of complex interactions between the intervention and the community system:

Conceptually, it (the assertion) frames health matters embedded in a community ecology that includes local conditions, community history, relations among subgroups in the community, relations between community groups and groups external to the community (including relationships with community intervention researchers), local resources, networks and their social capital, and effects of macrosystem policies on community life [24].

The authors recommend a shift from the traditional research paradigm in public health that is ‘intervention-centered’ to one that is ‘context-centered’ where research investigates community ecology, capacity and culture. South and Phillips build on this proposition in setting out some principles to guide the evaluation of community engagement and empowerment based on an understanding of communities as natural systems where the agency of community members can shape the intervention, the outcomes, and the wider public health system [12]. They argue that community engagement/empowerment interventions are rarely standardised, bounded public health interventions and therefore evaluation needs to account for complexity and the non-linear development processes in communities. Studies that have applied complex systems thinking to evaluate empowerment interventions include Durie and Wyatt’s case study of a community-led partnership where a complexity lens is used to analyse change processes [28] and a systems evaluation of Big Local, a UK area-based empowerment initiative, where Orton and colleagues argue that examining context with ethnographic methods offers a key to understanding how an empowerment intervention evolves [29].

This paper adds to this literature by discussing the evaluation approach used to capture the processes and impacts of Local People; a UK community empowerment intervention delivered with 29 neighbourhoods and communities of interest experiencing disadvantage. It explores how the design addressed the complexity within the programme and the varied nature of multiple community systems. A core argument is the value of case studies to illuminate the dynamic community contexts in which the intervention develops.

## 2. Background

‘Local People’ is a community empowerment programme that was designed to give people control over decisions leading to positive changes in personal or community conditions—environmental, social or economic—and ultimately to improve health and wellbeing [30]. Funded by the People’s Health Trust, a UK charity, each participating neighbourhood or community of interest has been granted between GBP 40,000 to GBP 85,000 per year through the programme to develop community skills, capacity and activities. Five national charities (referred to as partner organisations) supported the local implementation of the programme, each facilitating a group of communities to set local priorities and then supporting residents to develop a local project and take collective action. The intervention has been predominately targeted at neighbourhoods in the third most deprived areas of the country (based on the government’s Index of Multiple Deprivation); however, in some instances the focus has been on a community of interest within a wider area. The programme goals, set out by the People’s Health Trust in a theory of change (ToC), were to strengthen social connections and increase the control that people have individually and collectively over their lives and neighbourhoods [30]. Evidence suggests that empowerment at both an individual level and collectively in communities is associated with better health outcomes [31,32] through a set of interlinked pathways that attempt to break the link between powerlessness, deprivation and poor health [15,33]. The evaluation offered an opportunity to apply some of the principles for evaluating community engagement using a systems lens proposed by South and Phillips [12]. The Local People programme presented specific challenges for evaluation due to potential variation between the different community contexts. It was hypothesised that change pathways would be complex, non-linear and more critically, develop in community rather than professional systems in response to local needs and capacities [12,28].

## 3. Evaluation Design & Methods

The evaluation aimed to provide a formative and summative account of the development, implementation and intermediate impacts of the Local People programme and contribute to the evidence base around the effectiveness of community empowerment programmes in addressing the social determinants of health. Two research teams, one a university team and one based in a national think tank, worked jointly on the evaluation. Shared principles were agreed for the conduct of the evaluation: adopting a pragmatic approach to yield the best possible evidence to support programme development and aligning the evaluation to the ambition for greater collective control, including creating opportunities for the participation of local organisations and community members in the evaluation. The two teams worked to develop a joint ethical statement which described in detail the ethical approach for the evaluation as a piece of community-based research. Informed by the Social Research Association ethical guidelines [34], appropriate ethical safeguards around consent, anonymity, participant safety and risk reduction were then put in place. Ethical approval was given by Leeds Beckett University.

A mixed method design was used to give breadth and depth in the assessment of processes, impacts and outcomes at both programme and community project levels [35]. The ambition was to apply a systems lens throughout the evaluation, recognising the importance of community systems (context, culture and networks) [24] and change processes that lead to greater individual and community agency [12]. The pre-existing programme theory of change (ToC) was applied as a framework to guide the evaluation design, data collection and synthesis [36], and research questions were developed and then mapped to the ToC (see Appendix A). There was also a strand of self-evaluation where participating communities were given training and support to self-evaluate their progress and capture community-determined outcomes. Figure 1 shows the overall design and multiple data sources covering the different programme levels and stakeholder perspectives.

The evaluation used both survey and case study methods to yield both quantitative and qualitative data. The emphasis was on collecting data at a community-level and across intervention areas. The main components of the design were:(a)Case studies of Local People intervention communities: The case studies formed a major part of the qualitative fieldwork within the overall evaluation with the aim of providing an in-depth analysis of how the initiative worked in and for communities experiencing disadvantage (neighbourhoods and communities of interest). Using a multiple case study design [37], five case studies were undertaken in communities receiving Local People funding. The sampling strategy aimed to select a varied sample of areas, taking into account geographical spread, type of area (urban, rural and coastal) and the supporting national partner organisation. Sites were excluded where there had been major delays with activities, although key informant interviews were later undertaken with a sample of projects in this category. Drawing on traditions of qualitative, naturalistic enquiry [38], the primary research methods were semi-structured individual interviews and group interviews with project staff, local partner representatives and community members, including those involved and not involved in Local People activities. Data collection took place in two phases, with the research team undertaking a number of visits to each community over the course of the project. In the first phase, researchers attended meetings, community events or went on resident-led walks to familiarise themselves with the area [39]. The sample for each case study was selected through discussion with the project leads and following this orientation period. Individuals were invited to participate in the study and with consent, all interviews and focus groups were digitally recorded. Researchers also took field notes of their observation and reflexive memos.(b)Peer research: Each case study site also undertook some peer (participatory) research [40] supported by the research team. The peer research element aligned to the empowerment goals of the Local People programme [30]. The aims were to build local evaluation skills and capacity and to work collaboratively with residents who wished to collect data on the reach and impact of the case study projects. In each area, a preparatory workshop was held and between four and ten local people were recruited who later undertook peer research. These peer researchers decided the focus of the data collection, in most instances the reach of the initiative and barriers to involvement, and then did a small number of informal interviews in the community. Peer researchers discussed their findings at a final workshop run by the research team. In one case study site, an external exhibition was held.(c)A longitudinal survey: A questionnaire-based survey of project participants and residents was undertaken across all 29 neighbourhoods (or projects with communities of interest) receiving funding. Projects used a self-completion questionnaire that was administered at four points in the programme. Questions covered demographic variables, type of participation in the local project, views of the neighbourhood and questions about the shorter-term individual outcomes, including confidence, social connectedness, knowledge and skills. The questionnaire drew on verified questions from existing national surveys, mostly the Community Life survey [41]. All project leads were contacted and encouraged to distribute the questionnaire to residents who had some involvement in the programme or programme activities. Questionnaires were administered through paper copies or online if preferred.(d)Process appraisal: A further strand of the evaluation was the process appraisal. This focused on understanding how Local People worked through the different levels of the programme: funder, national organisation, local project support and communities. In addition to data gathered through case studies, key informant interviews were conducted with national and local leads working in the five national partner organisations.

### Analysis and Synthesis

Data analysis was carried out using appropriate methods for each component of the evaluation, with interim analysis of the longitudinal survey data and case studies at each point of data collection. Qualitative data from the case studies, including peer researchers’ findings and the process appraisal, were coded thematically and the process was managed using NVIVO software to help with systematic coding and organisation. A chart of themes was developed [42] to reflect the programme theory of change, but with flexibility to uncover unexpected outcomes. Individual case reports were produced that triangulated multiple data sources for each case study using the common framework [37]. In the first phase, themes on the context, starting points and project aims were summarised in an interim narrative account. The findings on context presented in this paper are drawn from this first stage of analysis. The final stage involved cross-case analysis to gain an understanding of all cross cutting themes, common change mechanisms and differences using a matrix to plot findings [43].

Quantitative survey data were analysed for each wave (1–4) using STATA statistical software [44] to produce descriptive statistics of change over time across the key variables. Regression and multilevel modelling techniques were used to detect statistically significant changes, controlling for demographic and socio-economic variables. Further detail on quantitative analysis and results are presented elsewhere [45].

The final stage of the evaluation was a synthesis of findings from all data sources using the research questions as a framework (Appendix A). The research team held two face-to-face meetings to develop the synthesis, where the main findings, outliers and fit to the original ToC were discussed. Findings were summarised in a table, highlighting areas where evidence was well supported by data and where there was less certainty or gaps in evidence. These data summaries were used to produce the final narrative account and to draw conclusions that could be verified by evidence see [45]. 

## 4. Results—Understanding Study Contexts

An important part of applying a system lens in the evaluation involved examining the social context of the case study areas [37] and, given the goals of the Local People programme to reduce inequalities, exploring how disadvantage, and also positive aspects of community life, played out. This was a particular focus in the first phase of data collection where researchers asked open questions to elicit rich descriptions of the community, although the importance of context also emerged in later interviews. This section presents qualitative findings from the case studies that illustrate the significance of social context drawn from the first phase of data collection and analysis.

Of the five selected case studies, four centred on neighbourhoods defined as socioeconomically disadvantaged and one across a larger urban area, with a focus on working with a community of people with disabilities and their carers (Table 1). In total 76 people took part in the interviews and focus groups in Phase 1: Case study A (10), B (16), C (21), D (19), E (10). Within case analysis generated a set of interrelated descriptive and interpretive themes, demonstrating variation between study contexts. Cross-case analysis led to higher order themes on context in terms of (i) the place and perceived impacts of wider socioeconomic conditions, (ii) community infrastructure and the social connections between community members (iii) experience of spatial stigma and lack of influence alongside the existence of community strengths. These themes are reported in turn.


*(i) The place and wider socioeconomic conditions*


Across all case studies, interviewees spoke of positive aspects of life in the area, often related to green or blue space or social activities, but also of the factors, such as transport or poverty, that limited access:
There’s lot to link in with on a good day. The bad side of it I think is it depends on how much money you’ve got or how ill you are or what the nature of the particular illness is—it can be really hard to access those things. It kind of changes from being an amazing place to a place that’s a bit shit. (Case study E)

Local knowledge gave a sense of history and how views of a place were formed. For example, in Case study B, a strip of green space between two housing estates and the history of community land use in that space was relevant to the later development of community action. In case study E, a reported history of tolerance in the town contrasted with the barriers to access the beach, which later became the focus of a community-led campaign.

The impact of wider socioeconomic conditions on the place and people was a key theme across all case study areas. In describing community life, interviewees talked of the impact of poverty, austerity and post-industrial change. Alongside these structural factors were personal accounts of the negative impacts of poverty, which affected the ability to participate:
When schools broke up for holiday this year the following week the number of people in food banks almost doubled. It’s because the kids are having breakfast club and lunch [in school]. People are really living in poverty and can’t afford to even feed their children. It is so expensive to live these days; you have nurses using food banks. People have got a good salary but by the time you run your house, you pay your bills and everything, you’ve got nothing left over to buy bloody food. (Case study C)


*(ii) Community infrastructure and social connections*


Community infrastructure, that is the specific mix of services, local amenities and existing groups in each area, influenced the development of the programme activity. Locally situated services and places to meet were valued, but there were identified gaps, which differentially affected some sections of the community more than others:
Both areas don’t have a community centre, as such. They have a library and they have lots of churches, but they don’t have a thriving community hub where people can go to meet and go to groups and catch up with the people. (Case study B)

Community infrastructure was not of course a fixed state and recent cuts to services in response to government-driven austerity were highlighted in most case studies, including in Case study E where the individual effects of welfare reform on people with disabilities was an important issue. Against the background of socioeconomic disadvantage in all areas, some interviewees spoke of a history of collective action in their area and of the presence of community activists. Barriers to wider community involvement that resulted in a lack of influence were also reported.

The strength of social relationships in the community was a major theme, with interpretations of community connectedness differing between and within areas. Strong bonds between community members were reported and also instances of division, isolation or lack of trust:
We can be on one estate and say, ‘oh, you know just over the road…’ ‘No, no, I’m not crossing the road, we’re not crossing that bridge”. (Case study A)
Respondent 1: “The people are friendly. Everyone knows everyone. ”Respondent 2: “You find that with the poorer communities though. In essence they are closer and more connected.” (Case study C)

A historical perspective was often given. Social ties were described as strong in two areas, with many families knowing each other through generations. Yet social context could change as community ties weakened by movement of people in and out of the area as well as work demands:
“Community spirit is probably at an all time low”. (Case study A)
“…the communities here, they’re ex-mining communities, I think people have grown up and moved on and people have found themselves lonely and, you know, within their own community lonely and I think there’s something lacking in terms of how do we bring those people together.” (Case study D)


*(iii) Perceptions of stigma and community strengths*


Understanding community context meant exploring the interplay between different aspects of community life. Two cross cutting themes were around experience of stigma and unseen assets in terms of social bonds. In all the case studies, interviewees described how their community was viewed in a negative light, defined by features such as poverty and crime:
I think we do tend to be very defensive of the area because we are so used to our reputation. Wherever you go, people say things like ‘don’t go up there the wheels will be stolen off your car’. (Case study C)

Negative views of a community affected collective agency and across all case studies, the lack of influence with local services and decision makers was a theme. Criticisms of local authority responsiveness to needs and duties, especially concerning housing and green space, were voiced by some interviewees:
We have got […] just down the road, big city, beautiful city, a lot of the resources go to that and since we became an authority … we all feel very strongly but […] takes all the money and we are left with the crumbs and anything that’s rubbish. (Case study A)

Perceptions of spatial stigma were often contrasted to the more positive experience of community connections and mutual aid within a community:
So there’s this great sense of connected and belonging, I think, despite all the pressures and despite all the, whatever poverty means, despite all those, sort of, economic pressures, there is a sense of community. (Case study D)

Overall, the varied study contexts, which reflected specific community needs and strengths, influenced the development of local priorities, against a backdrop of socioeconomic disadvantage present in all areas (see Table 1). For example, in Case study B, the focus was on improving community spaces and social connectedness where there had historically been two separate communities, whereas in Case study C, the focus was on widening community engagement in local activities, which in turn built on existing social networks and groups. Features of the social context could shape the community response both in terms of challenges to address and pathways to positive change:
“I think there’s a barrier in-between our older generation and our younger generation, if I’m honest. I think the older generation see these younger ones as trouble makers, some of them, but when they get to know them they’re lovely, and that’s how it is.” (Case study D)“So in a way we had a strong community and people will get involved if there is an issue. If there is issues they’ll come and get involved but if there is no issue they’re just quite happy to go along you know.“ (Case study A)

## 5. Discussion

There are strong arguments to use a systems lens in public health research [2], and as the science develops [7], it is important that researchers are transparent about methods and share learning [10]. Systems science in public health covers a range of research positions, methodologies and methods [7] and different types of systems evaluations are used in practice [46]. This paper contributes to the debates over how to improve the evaluation of community-based interventions in ways that take account of community systems and ecologies [12,13,23,24]. Local People was a large scale complex community initiative [47] and the choice of a mixed method design was well justified as it produced a comprehensive picture of the programme and the perspectives of different stakeholders. Along with other community-based interventions, the challenges were the need to account for contexts, cultures and the differential impacts of community agency on the system and the intervention [12,24]. What made this particularly challenging was how the empowerment intervention was relatively unstructured as communities decided the shape and goals of the intervention in their area according to their understanding of local needs and assets (Table 1).

The study design was informed by systems thinking in taking a multi-level, multi-site view of an empowerment programme and through using qualitative research to examine the dynamic nature of community systems and responses [46]. The focus of this paper is on design choices rather than on evidence of impacts, although we recognise that design, programme theory and results ultimately need to be integrated into a whole system evaluation. The findings presented here confirm the importance of studying community ecology in evaluations of community engagement and empowerment [24]. The use of longitudinal qualitative research to examine local contexts is in keeping with other UK community empowerment evaluations [28,29,48]. The fieldwork drew on naturalistic traditions of qualitative research [38], however we recognise that incorporating ethnographic methods, with research teams working for longer times within the study communities, may have led to better more nuanced understandings of the community systems [48]. Although the design attempted to draw on multiple data sources to strengthen evidence, it did not include comparator (non-intervention) communities. Without a control or comparison group who were not exposed to the intervention, and follow-up of a cohort over time (rather than a ‘panel’ design), potential effects of other influences cannot be accounted for. Notwithstanding the value of experimental studies in assessing causality, they offer a poor fit with most community empowerment interventions [13]. In this evaluation, pragmatic design choices were in keeping with the varied contexts for implementation, however the limitations of taking a systems-led, multi-method approach are acknowledged.

Context was critical to understanding Local People as a relatively unbounded intervention operating in the liminal space between civil society and public agencies [49]. The challenge was to balance the contextual, qualitative data, thereby avoiding context stripping [25], with the logic of the enquiry that aimed to answer questions at a more abstracted level—e.g., does the programme work? This tension was the subject of many research team discussions in analysis, synthesis and reporting stages. As researchers, we often moved between relating the specifics of a place or individual narratives and discussing overall programme outcomes. That the ‘system’ was not a professional system with a common language of partnership made the higher levels of abstraction more difficult. The importance of managing the tension between the generation of ‘thick descriptions’ of community contexts and building a more general programme theory is also noted in an account of a multi-site ethnographic evaluation of an area-based community empowerment initiative [48]. The three critical elements that helped manage this tension in our study were the application of a theory of change, the centrality of case studies and keeping community (lay) perspectives central to interpretations.

The use of ToC is recommended for complex public health interventions in order to articulate the interrelationships between the different activities, mechanisms, and outcomes [36]. In this evaluation, the existing ToC developed by the funder offered an initial framework to investigate the complex empowerment pathways that were intended to deliver improvements in community conditions. In later stages, critical discussion based on this framework was an important part of synthesis. This notion of a ToC as a heuristic tool to effect better understanding of complex dynamic systems is in line with the early literature on Complex Community Initiatives where ToC was first developed [47]. Blamey and Mackenzie caution that testing of programme theories of change can be relatively superficial compared to Realistic Evaluation [50]. Although findings were mapped against the ToC and evidence statements developed, a realist approach may have produced a more in-depth analysis of programme theory [50].

The second significant aspect of design was the centrality of case studies. Case studies have value in studying community empowerment initiatives as the design can attend to social context and to the evolution of an intervention in community systems, especially if longitudinal data are collected [37]. The findings on context, which included geography, culture, history, community organisations and relationships in the five case study sites, were critical to emerging interpretations. Recognised cross case analysis techniques allowed comparison between sites and for patterns to be plotted [43]. Like Orton and colleagues [29], the research teams concluded that context was not merely a backdrop, but was a major factor in how the empowerment programme developed in local areas (Table 1). Moreover, using open questions about what it was like to live or work in a community revealed the impact of structural factors on community life. While this is unsurprising given what is known about the drivers of health inequalities [33], these findings confirm the value of studying the system as a whole when examining how a public health intervention works in disadvantaged neighbourhoods.

A further theme was the perceived stigmatisation of areas labelled as ‘bad’. This supports the argument that spatial stigma is an under-researched issue which may have wider equity impacts [51]. Conversely, the case study findings revealed community strengths and capacities which could be drawn on in responding to local priorities (Table 1). The heterogeneity between case study sites illustrated how starting points for developing social action were markedly different depending on community history, challenges and existing infrastructure. Overall, learning from this study emphasises the value of case study design in exploring the critical links between context, participants and intervention. It also supports the need for further robust qualitative studies to understand the development and impacts of community empowerment interventions [28,29,48].

The final point of learning is the importance of lay perspectives. This study highlights the value of participatory (peer) research in illuminating the perspectives of those not directly involved in the intervention. In this study, seeing the intervention as an event in a wider system [9] helped unpack the nature of participation and non-participation [52]. Gathering community (lay) knowledge on the wider social context also improved understanding of local priorities within a national community empowerment programme. There is a long tradition of Community Based Participatory Research (CBPR) in public health and health promotion [39,53] and application of these methods are recommended when using a systems/ecological framework [24]. Although this paper makes no claims to using novel techniques, our reflection is that participatory methods are of value in developing understanding of social processes which are outside of the intervention yet still within the community system.

Working within a paradigm that acknowledges the role of community context and the capacity within communities should not mean abandoning the aspiration to create a more robust evidence base. This paper identifies actions that can be taken by those engaged in community-based evaluation to bridge the gap between the practice of community empowerment, a core process for health promotion [15], and the construction of an evidence base [13]. Recent reviews of measuring empowerment and community resilience at a national level, developed through WHO Regional Office for Europe, conclude that policy makers should consider mixed methods designs and can gain valuable information from qualitative case studies that develop knowledge with marginalised communities as a supplement to quantitative data [17,54]. This study provides further evidence that understanding social context is vital for building public health action particularly where communities live in disadvantaged circumstances. The implications for practice-based evaluation in public health and health promotion include developing of a theory of change to explain programme linkages, gathering qualitative data on social context, incorporating community and stakeholder perspectives and reporting on community assets and strengths as well as needs. Taking systems lens in community-based research helps build evidence that fits with a whole-of-society, multisectoral approach to health [4,6].

## 6. Conclusions

Interest in complex systems thinking in public health offers opportunities to improve the evaluation of community engagement and empowerment interventions. The traditional research paradigm is ill suited to examining the complex pathways of empowerment processes and the role of community systems in shaping action. As the science of systems thinking begins to be applied to community-based research, it is important to share accounts of evaluation practice and to be transparent about design choices in response to complexity. This paper has discussed these choices in relation to the evaluation of a large scale empowerment programme based in disadvantaged neighbourhoods or working with communities of interest. The use of a multiple case study design was critical in investigating the varied social-cultural contexts for the intervention. Accounts of community life illuminated salient features such as the history of a place, changing relationships, the impact of wider determinants and the existence of stigma. Some practical actions for incorporating a systems lens at a community-level are proposed including building an understanding of social context into community-based public health evaluation. The conclusions are that community context should not be seen as a static backdrop to an intervention, but as a dynamic feature of a wider system. Future research on community-based public health interventions should consider the value of case studies that incorporate community perspectives both to avoid context stripping and to understand the system beyond the professional sphere.

## Figures and Tables

**Figure 1 ijerph-17-00091-f001:**
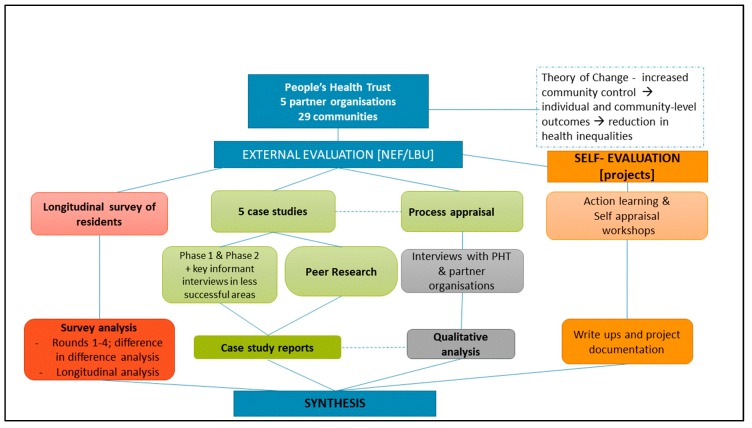
Local People evaluation design and data sources.

**Table 1 ijerph-17-00091-t001:** Summary of case study sites.

	Case Study Sites	Local Project Aims
A	Two neighbourhoods in the North West of England form the project area. It is an industrial area in decline, with high unemployment and low skilled jobs. Previously based around the docks, the focus of the area is now an out of town shopping centre. There are green spaces and good access to the countryside but a railway and a motorway divide the area. New people are arriving, but their stay is often short-term.	To improve the physical infrastructure of the area, enhance social spaces, make it more attractive and a better place to walk and cycle. Aimed at all residents living in the neighbourhoods. Supported by Sustrans.
B	On the outskirts of a large town in Scotland, two neighbourhoods form the project area. These are surrounded by green space and overlook the sea. Historically separate, the two neighbourhoods are linked by an area of woodland. Much of the current housing is former council owned, often terraced and a large new housing estate is being built. Wage stagnation and austerity has affected residents with many ‘just getting by’.	To get people involved in their community to help make it a better place ‘where they’re happier and more connected and where they feel they have a say’. Aimed at all residents living in the neighbourhoods. Supported by The Conservation Volunteers.
C	Two neighbourhoods form the project area. Set atop a hill in a city in Wales, the area has attractive views and convenient access to the city. Poverty rates are high with many residents having insufficient income. Food poverty is an issue and there are concerns for the future.	To support local residents to design and lead activities to improve the area for living and working in. Aimed at all residents living in the neighbourhoods. Supported by The Youth Sport Trust.
D	The project area is part of a small town in the West Midlands. The area is semi-rural. Housing is good quality and includes a former mining estate. Industries have shut and unemployment is high. There are limited opportunities for young people and reported low aspirations. The population is largely White British and with a high proportion of older people.	To improve health and wellbeing by connecting people back into community life, to be part of a society and to have a voice. The original target audience was 50 years plus, but this was broadened slightly to more inter-generational work. Supported by Royal Voluntary Service.
E	The project area is a busy, vibrant town on the South coast. Most of the town is not economically disadvantaged but there are pockets of deprivation. While there are liberal attitudes to diversity, there is some reported intolerance towards disability and accessibility to some areas is poor.	To create change, improve quality of life, to take control and make decisions. People with disabilities and unpaid carers are the communities of interest. The project is covers the town. Supported by Scope.

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
