# Peer review of "Complexity and Community Context: Learning from the Evaluation Design of a National Community Empowerment Programme"

_ijerph, 2019, doi:10.3390/ijerph17010091_

Round 1

Reviewer 1 Report

Thanks for the opportunity to review this paper.  It raises a number of key point about the limitations of traditional designs in the context of community empowerment evaluations and offers useful insights about how these were addressed in the evaluation of a place-based programme.

Points for consideration

The results section details the significance of social context for local communities and provides examples of how structural and local conditions affected the conditions in which people live and work.  Can the findings or discussion draw out some more detail about the significance of this context for understanding the way in which local projects developed as well as the changes that were observed as a result of the programme?  While some reference is made to this in the findings, a richer account would strengthen the arguments presented in the discussion. 

Results lines 253-258: The paragraph refers to both 'community infrastructure' and 'collective action' but it was unclear how these points were linked.  There also seems to be text missing at the end of the paragraph.

The first point in the discussion (Line 294 onwards) refers to the arguments in favour of systems approaches.  Consider spelling out more explicitly that there is often confusion about systems approaches (which are interpreted and used in a multitude of ways) hence the need for researchers to be transparent about methods and share learning.  This recent guide is one example of an attempt to improve clarity about what a systems approach is.

https://sphr.nihr.ac.uk/research/developing-a-systems-perspective-for-the-evaluation-of-local-public-health-interventions-theory-methods-and-practice/

Finally, this recent output also reflects on the role of qualitative approaches in evaluating an area based empowerment initiative and should be considered in the discussion.

Orton, L. et al (2019). Capturing complexity in the evaluation of a major area-based initiative in community empowerment: what can a multi-site, multi team, ethnographic approach offer? Anthropology and Medicine26(1), 48 64. https://doi.org/10.1080/13648470.2018.1508639

Title – it may also be worth revising the title to avoid confusion with this paper above?

Author Response

Thanks to Reviewer 1 for their helpful comments which we have used to draw out the discussion and link to further literature. In response to points:

·        The focus of this paper was on design choices in relation to dynamic community contexts rather than examining how context influenced local projects and outcomes. We were mindful that this latter subject is explored in some depth in a paper by Orton et al. (2016), which we reference and include in the introduction (line 82-86). Nevertheless, we have responded to these comments and included a short section in the findings about how context shaped the local priorities for the programme and linked this to Table 1 (Line 293-305). We have also included more in the first paragraph of the discussion (line 310-313) to recognise the significance of context on both programme/intervention and evaluation design.

·        Results lines 254-259.  We have amended this paragraph to make it clearer and removed the extra text.

·        In response to the suggestion that we consider spelling out more explicitly that there is often confusion about systems approaches (which are interpreted and used in a multitude of ways). We have amended the first paragraph of the discussion (Lines 311-314) and referenced the suggested report. We have made small changes throughout the discussion to locate our paper in wider debates but make clear our specific focus on lines 322 and 329-30.

·        In light of the comment on qualitative research and the suggestion for referencing the recent paper, we have amended the discussion to give further emphasis to points about the value of qualitative case studies in evaluating an empowerment initiative (eg lines 382-83). This is also picked up in the new final paragraph on implications for practice. (lines 395-410)

·        The Orton et al. (2019) paper is now referenced and their helpful discussion about balancing the tensions is now referred to in the discussion (lines 326-330;347-349)

·        Title – it may also be worth revising the title to avoid confusion with this paper above? We have revised the title. We think this is very sensible given the potential for confusion.  Apologies for this as we had not seen this recent paper.

Reviewer 2 Report

This paper does what is says it will do: provides a description of design choices and tensions from the evaluation of a national community empowerment programme. It is an interesting paper written by researchers for the benefit of other researchers and therefore has a limited readership appeal.

What is missing is a discussion of the practice implications for the design and evaluation of future public health programmes. This will require using a different style of writing, one that is less academic and rigid, and one that can be alsounderstood by programme planners and policy makers.

The authors may wish to look at the excellent WHO discussion of the practice implications for the national measurement of community empowerment / resilience as a guide to a review of this paper: Health Evidence Network (HEN) synthesis report 59 & 60. 2018, World Health Organisation. Regional Office for Europe: Copenhagen.

Author Response

Thanks to the reviewer for the recommendation to include a discussion of the practice implications for the design and evaluation of future public health programmes. We have now added a new paragraph at the end of the Discussion about the practice implications for evaluation (line 395-410). This is a research paper reporting on an academic study; however, we agree that there are some points of learning for practice which hopefully this additional paragraph highlights. The 2 reports (one of which I authored) are now referenced. We also refer back to need for evaluation that fits with multisectoral action which is discussed in the Introduction.

 In addition to responding to reviewers’ comments, we have also added a missing reference and edited a couple of inconsistent phrases and typos.

Round 2

Reviewer 2 Report

I would like to reiterate my previous comments: 'What is missing is a discussion of the practice implications for the design and evaluation of future public health programmes. This will require using a different style of writing, one that is less academic and rigid, and one that can be also understood by programme planners and policy makers'.

The additional comments by the authors are vague, in general and do not add anything new-choosing to state that the findings support what is already know. What is new about these findings and what are the practical implications. This paragraph should be expanded, made specific and moved to the conclusions.

Author Response

Thank you for these comments. We have moved the main recommendation for evaluation into the conclusion.